# Heavy metal exposure and its impact on inflammatory ratios in minors: The mediating role of BMI

Xinpeng Li🆔, Lu Han🆔 *

Department of Clinical Laboratory, Public health clinical center of Chengdu, Chengdu, China

* boumaduoduo@163.com

## Abstract

### Background

Despite existing evidence that endocrine-disrupting chemicals like heavy metals exposure impairs health of minors, the association between the exposures and inflammatory ratios remains uncertain. This study aims to investigate the relationship between heavy metal exposure and inflammatory ratios, focusing on BMI as a potential mediator in this association.

### Method

We conducted a retrospective cross-sectional analysis from the NHANES 2007–2018. 14,007 minors were categorized into different age groups, and analyses were performed based on demographic characteristics. Multiple linear regression and mediation analysis were applied to asses associations between heavy metal concentrations and inflammatory ratios, with BMI included as a mediating variable.

### Results

The participants were divided into four age groups: toddlers (2487), preschool children (2297), school-age children (5019), and teenagers (4204). Blood Pb was positively correlated with LMR ($\beta = 0.70$, 95% CI: 0.60–0.81) and PNR ($\beta = 14.88$, 95% CI: 12.29–17.47), with 25.89% and 27.02% of these associations mediated by BMI. Negative correlations were observed between Pb and inflammation ratios, including NLR ($\beta = -0.29$, 95% CI: $-0.34 - -0.24$), PLR ($\beta = -10.35$, 95% CI: $-12.61 - -8.08$), and NMR ($\beta = -0.63$, 95% CI: $-0.78 - -0.48$), with BMI accounting for 37.64%, 22.40%, and 39.59% of these effects, respectively. Blood Cd and Hg were also correlated with these ratios, with BMI consistently mediating these associations.

**Data availability statement:** The original data-sets were obtained from the official NHANES website (https://wwwn.cdc.gov/nchs/nhanes/search/datapage.aspx?Component=Laboratory&CycleBeginYear=2017), and data from 2007 to 2018 were subsequently integrated for analysis. The raw data are available at the following link: https://doi.org/10.6084/m9.figshare.30353935.

**Funding:** The author(s) received no specific funding for this work.

**Competing interests:** The authors have declared that no competing interests exist.

**Abbreviations:** BMI, Body mass index; CBC, Complete blood count; Cd, Cadmium; EDCs, Endocrine-disrupting chemicals; Hg, Mercury; LC, Lymphocyte count; LMR, Lymphocyte-to-monocyte ratio; MC, Monocyte count; NC, Neutrophil count; NHANES, National Health and Nutrition Examination Survey; NLR, Neutrophil-to-lymphocyte ratio; NMR, Neutrophil-to-monocyte ratio; Pb ,Lead; PC, Platelet count; PLR, Platelet-to-lymphocyte ratio; PNR, Platelet-to-neutrophil ratio.

## Conclusions

BMI serves as a significant mediator between blood heavy metals and inflammatory ratios among minors.

---

## 1. Introduction

Endocrine-disrupting chemicals (EDCs) are environmental pollutants that interfere with the normal functioning of the endocrine system, potentially resulting in a wide range of metabolic disturbances [1]. Among them, heavy metals such as lead (Pb), cadmium (Cd), and mercury (Hg), poses a significant challenge to public health due to their widespread environmental presence and profound health impacts [2–5]. Chronic exposure to Pb has been associated with cognitive dysfunction, behavioral abnormalities, and developmental delays in children, as well as hypertension and renal impairment in adults [6]. The developing fetus and children are particularly vulnerable to the neurodevelopmental and neurobehavioral effects of Pb exposure [7]. As a potent industrial carcinogen, Cd poses a substantial risk to children and adolescents. Early-life exposure to Cd may lead to respiratory problems, dermatological lesions, and an increased risk of lung cancer, contributing to long-term developmental impairments [8]. Hg, primarily ingested through contaminated fish and seafood, is particularly hazardous in the form of methylmercury, a compound highly toxic to the nervous system and capable of causing neurodevelopmental deficits and systemic health problems in developing fetuses [9, 10]. These heavy metals present poses direct threats to the health of minors and have profound public health implications due to their persistence and bioaccumulative nature within ecosystems.

Evidence increasingly points to the clinical significance of hematological parameters derived from routine peripheral blood analyses as valuable inflammatory markers in a variety of clinical conditions. Key ratios include the neutrophil-to-lymphocyte ratio (NLR), lymphocyte-to-monocyte ratio (LMR), platelet-to-lymphocyte ratio (PLR), platelet-to-neutrophil ratio (PNR), and neutrophil-to-monocyte ratio (NMR) [11,12]. Although research on these markers has primarily focused on adult populations, particularly cancer patients [13,14], their potential role in minors is gaining increasing attention. Minors are in a developmental stage, with an immature immune system, making them more sensitive to external factors [1,15]. Studies have demonstrated that environmental exposures, particularly to heavy metals, are directly associated with elevated levels of inflammatory marker [16]. In developing children, such exposures may contribute to long-term health consequences, including chronic inflammation, immune dysfunction, and metabolic disorders [17]. Therefore, investigating alterations in inflammatory markers among minors could help identify high-risk individuals and provide a scientific basis for early intervention and disease prevention.

Emerging research indicates that the deleterious effects of heavy metal exposure are intricately linked to disruptions in immune homeostasis and the promotion of systemic inflammation. This indicates a complex interplay between heavy metal exposure and immune function in children [18,19]. Long-term exposure to heavy

metals can disrupt hematopoietic homeostasis by interfering with the generation, maturation, and functional of activity of lymphocytes, neutrophils, and monocytes. Such disturbances may result in altered proportions of these cellular subsets in systemic circulation. For example, elevated Cd exposure has been shown to increase NLR and PLR values—reflecting enhanced neutrophil-driven inflammation and platelet activation—while simultaneously decreasing LMR, indicative of relative lymphocytopenia and/or monocyte proliferation [20]. A study conducted in China assessing heavy metal concentrations and inflammatory biomarkers in polluted regions found that individuals residing in these areas exhibited higher NLR and PLR values, accompanied by lower LMR, compared with those from control regions. Moreover, the immune inflammatory response was further aggravated with increasing co-exposure to Cd and Pb. [21]. Elevated blood levels of Pb and Cd have been significantly associated with alterations in immune-inflammatory biomarkers in both exposed and control populations. These observations are consistent with findings from other studies, reinforcing the evidence that heavy metal exposure profoundly affects immune regulation and inflammatory pathways [22,23].

This study draws on data from the National Health and Nutrition Examination Survey (NHANES) to investigation the association between blood concentrations of heavy metals and inflammation-related hematological ratios, including NLR, LMR, PLR, PNR, and NMR. In particular, it examines the potential mediating role of body mass index (BMI), exploring whether prolonged exposure to heavy metals may contribute to chronic systemic inflammation through its influence on BMI.

## 2. Methods

### 2.1. Population

The study population was derived from the NHANES 2007–2018. Initially, 59842 participants were considered. Individuals with missing data on blood heavy metals (n = 17776) or peripheral blood cell counts (n = 5311) were excluded. Additional exclusions were applied to 22748 participants, including 24 with HIV infection, 166 who were pregnant or uncertain of their pregnancy status, and 22558 aged 18 years or older. A total of 14007 minors were finally included in the analysis (Fig 1).

### 2.2. Assessment of blood heavy metals

The exposure variables in this study included three heavy metals in blood: Pb, Cd, and Hg. The concentrations of these metals were measured using inductively coupled plasma mass spectrometry (ICP-MS) with quadrupole technology.

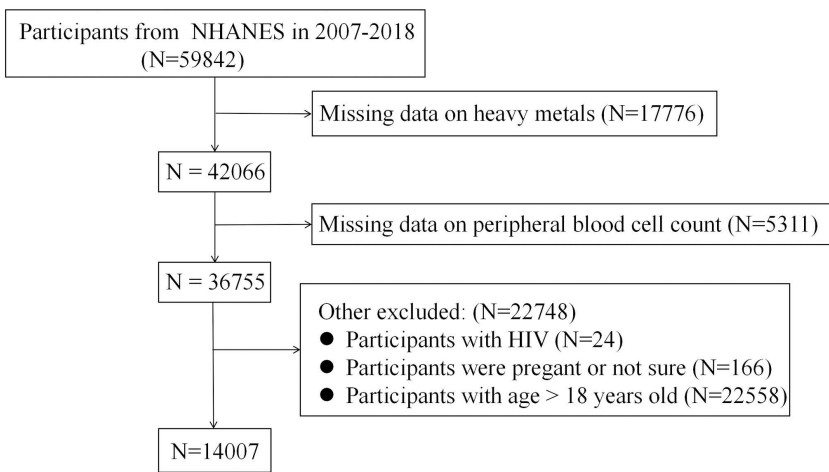

**Fig 1. Participant selection flowchart from NHANES database.**

## 2.3. Routine blood-related inflammatory markers

The ratios of NLR, LMR, PLR, PNR and NMR mainly relied on the results of the complete blood count (CBC) test. The CBC with 5-Part differential of NHANES was performed using the Beckman Coulter methodology. The test also included automatic dilution and mixing of samples, as well as hemoglobinometry performed using a single beam photometer. Additionally, the platelet count (PC), neutrophil count (NC), lymphocyte count (LC), and monocyte count (MC) were used to calculate these ratios. NLR is calculated as neutrophil-to-lymphocyte ratio, LMR is the lymphocyte-to-monocyte ratio, PLR is the platelet-to-lymphocyte ratio, PNR is the platelet-to-neutrophil ratio, and NMR is the neutrophil-to-monocyte ratio.

## 2.4. Covarietes

To examine children at various developmental stages, the study population was divided into four age groups: toddlers (< 3 years), pre-school children (4–6 years), school-age children (7–12 years), and teenagers (13–18 years). After applying weighting adjustments, the study population was stratified by race/ethnicity into the following groups: Mexican American and Hispanic, Non-Hispanic White and Multi-Racial, Non-Hispanic Black, and Non-Hispanic Asian. BMI was classified according to the World Health Organization (WHO) criteria as normal weight (BMI < 25), overweight (25 ≤ BMI < 30), and obesity (BMI ≥ 30). Information on diabetes and blood pressure status were obtained from self-reported questionnaire data.

## 2.5. Statistic analysis

Continuous variables were presented as mean ± standard deviation, whereas categorical variables were expressed as percentages. Spearman's correlation analysis was conducted to assess the relationships among all independent variables and covariates. Multivariate linear regression models with covariate adjustment were used to examine the associations between blood heavy metal concentrations and inflammatory ratios. The generalized additive model was applied to explore the nonlinear relationship between Hg and inflammatory markers and to perform curve fitting.

## 2.6. Mediation analysis

We further examined whether BMI mediated the associations between blood heavy metal concentrations (Cd, Pb, and Hg) and inflammatory markers (NLR, LMR, PLR, PNR, and NMR). The mediation analysis was performed using the "mediation" package in R version 4.5.1. In this model, blood heavy metal levels were treated as independent variables, inflammatory ratios as dependent variables, and BMI as the mediator. The total effect of heavy metals on inflammatory markers was divided into direct effects and indirect effects through BMI. The bootstrap method with 5,000 resamples was used to estimate the indirect effects and their 95% confidence intervals. A mediation effect was considered significant when the confidence interval did not include zero. The proportion of mediation was calculated as the percentage of the indirect effect relative to the total effect.

All statistical analyses were performed using R software (https://www.r-project.org) and EmpowerStats (https://www.empowerstats.net/cn/).

## 3. Results

### 3.1. Baseline

A total of 14007 participants were included in the analysis, with 2487 toddlers, 2297 preschool children, 5019 school-age children, and 4204 teenagers. Among these participants, 51.2% were male and 48.8% were female. According the BMI classification, 82.64% of participants had a BMI < 25, 10.51% between 25 and 30, and 6.84% above 30. However, with increasing age, the proportion of individuals classified as obese (BMI ≥ 30) gradually increased, reaching 14.88% in the teenager group (Table 1). A positive correlation between BMI and age was also observed (Fig 2, $r = 0.54$). Concurrently,

**Table 1. The basic demographic characteristics of participants.**

| | Total | Toddler (< 3) | Pre-School Child (4–6) | School-age Child (7–12) | Teanager (13–18) | P value |
|---|---|---|---|---|---|---|
| Number (N) | 14007 | 2487 | 2297 | 5019 | 4204 | |
| Gender, % (N) | | | | | | 0.6398 |
| Male | 51.20 (7172) | 51.34 (1277) | 51.54 (1184) | 51.76 (2598) | 50.53 (2124) | |
| Female | 48.80 (6835) | 48.66 (1210) | 48.46 (1113) | 48.24 (2421) | 49.47 (2080) | |
| Race/ethnicity, % (N) | | | | | | <0.0001 |
| Mexican American and Hispanic | 23.98 (3359) | 28.06 (698) | 27.50 (631) | 24.85 (1247) | 20.58 (865) | |
| Non-Hispanic White and Multi-Racial | 53.33 (7470) | 48.16 (1198) | 48.44 (1113) | 53.31 (2675) | 56.86 (2390) | |
| Non-Hispanic Black | 13.96 (1955) | 15.09 (375) | 15.01 (345) | 13.38 (672) | 13.70 (576) | |
| Non-Hispanic Asian | 8.73 (1223) | 8.70 (216) | 9.05 (208) | 8.46 (425) | 8.86 (373) | |
| BMI Group, % (N), kg/m2 | | | | | | <0.0001 |
| < 25 | 82.64 (11575) | 99.8 (2482) | 98.69 (2267) | 87.99 (4416) | 66.39 (2791) | |
| 25 ≤ BMI < 30 | 10.51 (1473) | 0.20 (5) | 1.23 (28) | 8.92 (448) | 18.73 (787) | |
| ≥ 30 | 6.84 (959) | 0.00 | 0.08 (2) | 3.09 (155) | 14.88 (626) | |
| PIR | 2.55 ± 1.69 | 2.27 ± 1.66 | 2.31 ± 1.69 | 2.52 ± 1.67 | 2.75 ± 1.68 | <0.0001 |
| Diabetes, % (N) | | | | | | <0.0001 |
| Yes | 0.43 (60) | 0.13 (3) | 0.05 (1) | 0.34 (17) | 0.74 (31) | |
| No | 99.16 (13889) | 99.75 (2481) | 99.89 (2294) | 99.18 (4987) | 98.67 (4148) | |
| Unknow | 0.42 (59) | 0.12 (3) | 0.06 (1) | 0.48 (24) | 0.59 (25) | |
| Blood Pressure, % (N) | | | | | | <0.0001 |
| Regular | 73.94 (10357) | 79.12 (1968) | 79.29 (1821) | 76.76 (3853) | 67.79 (2850) | |
| Irregular | 0.16 (22) | 0.28 (7) | 0.00 | 0.25 (13) | 0.10 (4) | |
| Unknow | 25.9 (3628) | 20.61 (513) | 20.71 (476) | 22.99 (1154) | 32.11 (1350) | |
| Vitamin D, nmol/L | 67.94 ± 20.82 | 73.35 ± 17.19 | 72.58 ± 17.76 | 68.31 ± 19.73 | 64.13 ± 23.01 | <0.0001 |
| NLR | 1.49 ± 0.96 | 0.81 ± 0.55 | 1.27 ± 1.04 | 1.46 ± 0.86 | 1.83 ± 0.98 | <0.0001 |
| PLR | 114.05 ± 43.01 | 85.43 ± 36.42 | 109.72 ± 46.88 | 118.34 ± 40.63 | 121.28 ± 41.51 | <0.0001 |
| LMR | 5.02 ± 2.07 | 6.76 ± 2.77 | 5.56 ± 2.05 | 4.92 ± 1.79 | 4.33 ± 1.60 | <0.0001 |
| PNR | 93.34 ± 51.69 | 129.76 ± 78.08 | 103.59 ± 51.21 | 93.93 ± 45.55 | 77.08 ± 36.77 | <0.0001 |
| NMR | 6.51 ± 2.85 | 4.78 ± 2.26 | 6.17 ± 2.62 | 6.52 ± 2.66 | 7.18 ± 3.01 | <0.0001 |
| Blood Cd, μg/L | 0.16 ± 0.18 | 0.12 ± 0.04 | 0.12 ± 0.05 | 0.14 ± 0.07 | 0.21 ± 0.27 | <0.0001 |
| Blood Pb, μg/dL | 0.88 ± 0.94 | 1.44 ± 1.73 | 1.10 ± 1.02 | 0.81 ± 0.62 | 0.68 ± 0.63 | <0.0001 |
| Blood Hg, μg/L | 0.56 ± 0.81 | 0.41 ± 0.91 | 0.48 ± 0.66 | 0.55 ± 0.72 | 0.65 ± 0.88 | <0.0001 |

P value for the continuous variables was calculated by weighted linear regression model.

P value for the categorical variables was calculated by weighted chi-square test.

vitamin D levels progressively decrease ($r = -0.31$). The concentrations of Cd and Hg increased with age, whereas Pb levels decreased. Additionally, the values of NLR, PLR and NMR showed an upward trend, while the LMR and PNR decreased with age. Specifically, Fig 2 shows that the correlations among all independent variables and covariates were relatively low, indicating that multicollinearity was not a major concern.

### 3.2. Association analysis between blood heavy metals and ratios

After adjusting for all covariates, NLR showed a significant increase with higher quartiles of Cd exposure, peaking in Q4 ($\beta = 0.26$, 95%CI: 0.22–0.31), whereas Pb exposure was associated with a significant decline in NLR, with the greatest reduction in Q4 ($\beta = -0.29$, 95% CI: −0.34–−0.24). A similar pattern was observed for PLR. Conversely, LMR decreased

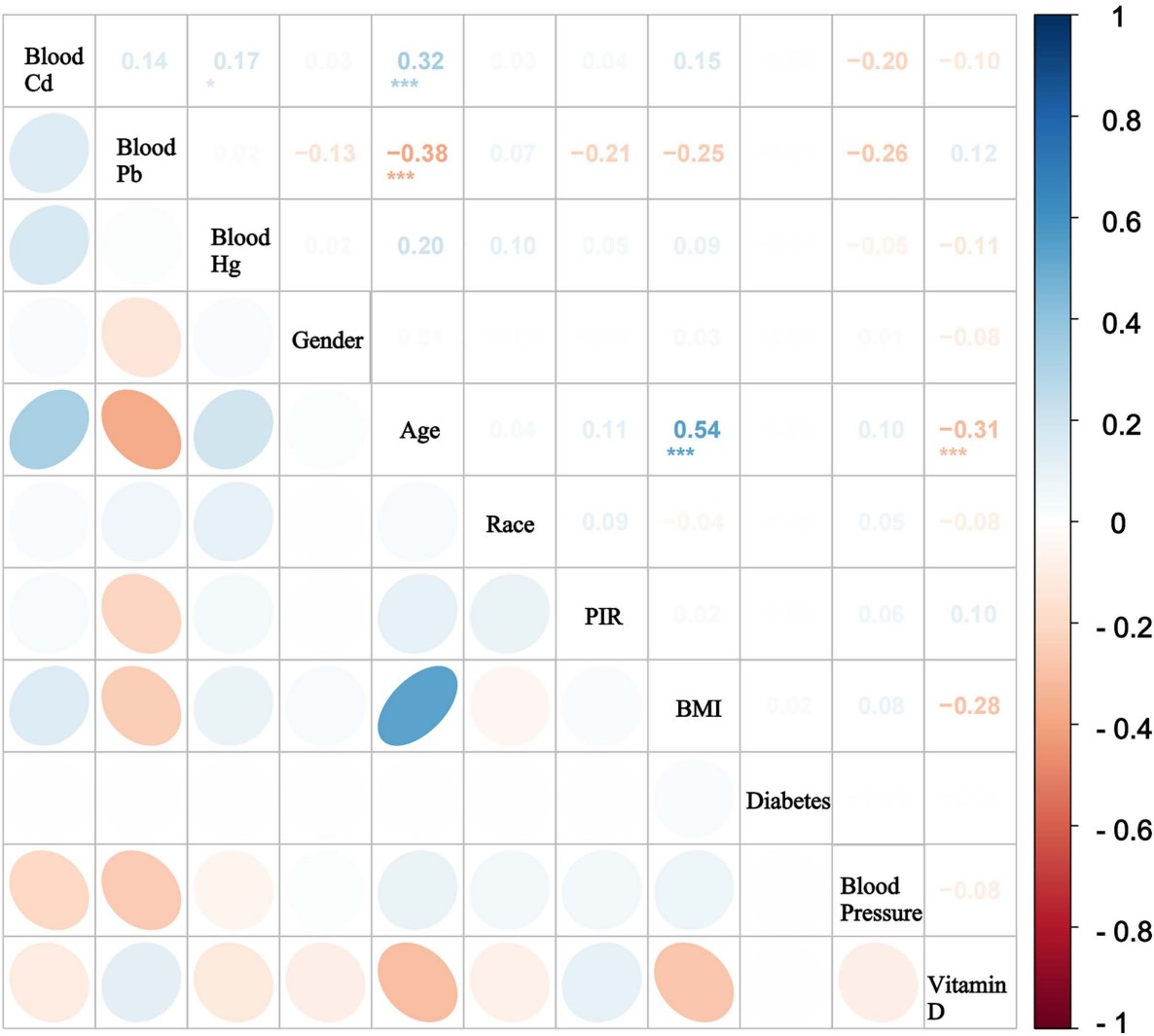

**Fig 2. Spearman's correlation analysis plot of variables.** ***: *P<0.001* **: *P<0.01* *: *P<0.05*.

significantly with increasing Cd exposure, particularly in Q4 (*β*=−0.58, 95% CI: −0.68—-0.48), while Pb exposure was associated with a marked increase in LMR, with the highest effect in Q4 (*β*=0.70, 95% CI: 0.60–0.81). PNR showed a substantial increase with higher quartiles of Pb exposure, with the most pronounced effect in Q4 (*β*=14.88, 95% CI: 12.29–17.47). Finally, NMR increased significantly with higher Cd exposure, particularly in Q4 (*β*=0.62, 95% CI: 0.48–0.76), and also rose with Pb exposure, reaching the highest coefficient in Q4 (*β*=0.63, 95% CI: 0.48–0.78). In contrast, Hg exposure did not show any significant linear association with these ratios (Table 2).

### 3.3. Exploring the nonlinear relationship between Hg and inflammatory markers

Given the absence of a significant linear relationship between Hg and inflammatory biomarkers, generalized additive models were applied to explore the dose–response relationships (Fig 3). The analysis revealed that NLR, NMR, and PLR exhibited relatively higher values at lower Hg concentrations but showed a decreasing trend as Hg levels increased. In

**Table 2. Multiple linear regression analysis between heavy metals and biomarkers.**

| Exposure | Non-adjusted | Adjust I | Adjust II |
|---|---|---|---|
| **NLR** | | | |
| Cd quartile (µg/L) | | | |
| Q1 (0.07, 0.11) | Ref. | Ref. | Ref. |
| Q2 (0.11, 0.14) | 0.16 (0.11, 0.21) <0.0001*** | 0.14 (0.09, 0.19) <0.0001*** | 0.14 (0.09, 0.19) <0.0001*** |
| Q3 (0.14, 0.16) | 0.19 (0.14, 0.23) <0.0001*** | 0.14 (0.10, 0.19) <0.0001*** | 0.14 (0.10, 0.19) <0.0001*** |
| Q4 (0.16, 6.09) | 0.34 (0.29, 0.38) <0.0001*** | 0.26 (0.22, 0.31) <0.0001*** | 0.26 (0.22, 0.31) <0.0001*** |
| Pb quartile (µg/dL) | | | |
| Q1 (0.05, 0.49) | Ref. | Ref. | Ref. |
| Q2 (0.49, 0.73) | −0.18 (−0.22, −0.14) <0.0001*** | −0.11 (−0.15, −0.07) <0.0001*** | −0.11 (−0.16, −0.07) <0.0001*** |
| Q3 (0.73, 1.15) | −0.26 (−0.31, −0.22) <0.0001*** | −0.15 (−0.19, −0.10) <0.0001*** | −0.16 (−0.21, −0.12) <0.0001*** |
| Q4 (1.15, 34.11) | −0.44 (−0.48, −0.39) <0.0001*** | −0.27 (−0.32, −0.22) <0.0001*** | −0.29 (−0.34, −0.24) <0.0001*** |
| Hg quartile (µg/L) | | | |
| Q1 (0.11, 0.20) | Ref. | Ref. | Ref. |
| Q2 (0.20, 0.34) | 0.01 (−0.07, 0.09) 0.8334 | 0.02 (−0.06, 0.09) 0.6924 | 0.02 (−0.06, 0.09) 0.6398 |
| Q3 (0.34, 0.62) | 0.08 (−0.00, 0.16) 0.0560 | 0.08 (−0.00, 0.16) 0.0545 | 0.08 (0.00, 0.16) 0.0494* |
| Q4 (0.62, 28.84) | 0.08 (0.00, 0.16) 0.0430* | 0.08 (0.00, 0.16) 0.0452* | 0.09 (0.01, 0.17) 0.0268* |
| **PLR** | | | |
| Cd quartile (µg/L) | | | |
| Q1 (0.07, 0.11) | Ref. | Ref. | Ref. |
| Q2 (0.11, 0.14) | 7.96 (5.64, 10.29) <0.0001*** | 7.57 (5.26, 9.89) <0.0001*** | 7.28 (4.91, 9.65) <0.0001*** |
| Q3 (0.14, 0.16) | 14.45 (12.44, 16.46) <0.0001*** | 14.04 (12.03, 16.05) <0.0001*** | 13.75 (11.60, 15.90) <0.0001*** |
| Q4 (0.16, 6.09) | 14.95 (12.88, 17.02) <0.0001*** | 13.93 (11.84, 16.03) <0.0001*** | 13.68 (11.54, 15.83) <0.0001*** |
| Pb quartile (µg/dL) | | | |
| Q1 (0.05, 0.49) | Ref. | Ref. | Ref. |
| Q2 (0.49, 0.73) | −5.01 (−6.90, −3.12) <0.0001*** | −4.74 (−6.65, −2.83) <0.0001*** | −4.99 (−6.92, −3.06) <0.0001*** |
| Q3 (0.73, 1.15) | −5.75 (−7.72, −3.77) <0.0001*** | −5.32 (−7.35, −3.30) <0.0001*** | −5.72 (−7.79, −3.66) <0.0001*** |
| Q4 (1.15, 34.11) | −10.14 (−12.25, −8.04) <0.0001*** | −9.67 (−11.86, −7.47) <0.0001*** | −10.35 (−12.61, −8.08) <0.0001*** |
| Hg quartile (µg/L) | | | |
| Q1 (0.11, 0.20) | Ref. | Ref. | Ref. |
| Q2 (0.20, 0.34) | 2.67 (−0.84, 6.18) 0.1355 | 2.46 (−1.04, 5.95) 0.1683 | 2.92 (−0.60, 6.44) 0.1043 |
| Q3 (0.34, 0.62) | 3.52 (−0.08, 7.13) 0.0553 | 2.91 (−0.69, 6.51) 0.1136 | 3.26 (−0.37, 6.88) 0.0781 |
| Q4 (0.62, 28.84) | 3.59 (−0.05, 7.22) 0.0530 | 3.12 (−0.52, 6.76) 0.0930 | 3.68 (0.02, 7.34) 0.0488* |
| **LMR** | | | |
| Cd quartile (µg/L) | | | |
| Q1 (0.07, 0.11) | Ref. | Ref. | Ref. |
| Q2 (0.11, 0.14) | −0.27 (−0.38, −0.16) <0.0001*** | −0.24 (−0.35, −0.13) <0.0001*** | −0.28 (−0.39, −0.17) <0.0001*** |
| Q3 (0.14, 0.16) | −0.50 (−0.60, −0.40) <0.0001*** | −0.42 (−0.51, −0.32) <0.0001*** | −0.48 (−0.58, −0.38) <0.0001*** |
| Q4 (0.16, 6.09) | −0.63 (−0.73, −0.53) <0.0001*** | −0.55 (−0.64, −0.45) <0.0001*** | −0.58 (−0.68, −0.48) <0.0001*** |
| Pb quartile (µg/dL) | | | |
| Q1 (0.05, 0.49) | Ref. | Ref. | Ref. |
| Q2 (0.49, 0.73) | 0.27 (0.18, 0.36) <0.0001*** | 0.22 (0.13, 0.31) <0.0001*** | 0.20 (0.11, 0.29) <0.0001*** |
| Q3 (0.73, 1.15) | 0.45 (0.35, 0.54) <0.0001*** | 0.33 (0.24, 0.42) <0.0001*** | 0.32 (0.22, 0.41) <0.0001*** |
| Q4 (1.15, 34.11) | 0.91 (0.81, 1.01) <0.0001*** | 0.71 (0.61, 0.82) <0.0001*** | 0.70 (0.60, 0.81) <0.0001*** |
| Hg quartile (µg/L) | | | |
| Q1 (0.11, 0.20) | Ref. | Ref. | Ref. |
| Q2 (0.20, 0.34) | −0.18 (−0.34, −0.01) 0.0397* | −0.21 (−0.37, −0.04) 0.0139* | −0.18 (−0.34, −0.01) 0.0353* |

*(Continued)*

**Table 2.** (Continued)

| Exposure | Non-adjusted | Adjust I | Adjust II |
|---|---|---|---|
| Q3 (0.34, 0.62) | −0.32 (−0.50, −0.15) 0.0002*** | −0.38 (−0.55, −0.21) <0.0001*** | −0.35 (−0.52, −0.18) <0.0001*** |
| Q4 (0.62, 28.84) | −0.32 (−0.50, −0.15) 0.0003*** | −0.40 (−0.57, −0.23) <0.0001*** | −0.39 (−0.56, −0.22) <0.0001*** |
| **PNR** | | | |
| **Cd quartile (µg/L)** | | | |
| Q1 (0.07, 0.11) | Ref. | Ref. | Ref. |
| Q2 (0.11, 0.14) | −3.98 (−6.74, −1.22) 0.0047** | −3.58 (−6.22, −0.94) 0.0079** | −4.32 (−7.03, −1.62) 0.0017** |
| Q3 (0.14, 0.16) | −0.46 (−2.85, 1.93) 0.7075 | 2.51 (0.22, 4.81) 0.0319* | 1.25 (−1.20, 3.70) 0.3183 |
| Q4 (0.16, 6.09) | −9.43 (−11.89, −6.96) <0.0001*** | −5.68 (−8.07, −3.29) <0.0001*** | −6.44 (−8.89, −3.99) <0.0001*** |
| **Pb quartile (µg/dL)** | | | |
| Q1 (0.05, 0.49) | Ref. | Ref. | Ref. |
| Q2 (0.49, 0.73) | 7.45 (5.20, 9.69) <0.0001*** | 2.69 (0.51, 4.87) 0.0154* | 2.47 (0.27, 4.68) 0.0278* |
| Q3 (0.73, 1.15) | 14.18 (11.83, 16.53) <0.0001*** | 6.49 (4.18, 8.79) <0.0001*** | 6.32 (3.96, 8.67) <0.0001*** |
| Q4 (1.15, 34.11) | 26.29 (23.79, 28.79) <0.0001*** | 14.90 (12.39, 17.40) <0.0001*** | 14.88 (12.29, 17.47) <0.0001*** |
| **Hg quartile (µg/L)** | | | |
| Q1 (0.11, 0.20) | Ref. | Ref. | Ref. |
| Q2 (0.20, 0.34) | 3.29 (−0.88, 7.46) 0.1219 | 2.20 (−1.79, 6.19) 0.2791 | 2.85 (−1.17, 6.87) 0.1649 |
| Q3 (0.34, 0.62) | −1.76 (−6.04, 2.53) 0.4220 | −3.31 (−7.42, 0.80) 0.1140 | −2.70 (−6.83, 1.44) 0.2015 |
| Q4 (0.62, 28.84) | −4.09 (−8.41, 0.23) 0.0633 | −5.44 (−9.59, −1.29) 0.0103* | −5.08 (−9.26, −0.90) 0.0171* |
| **NMR** | | | |
| **Cd quartile (µg/L)** | | | |
| Q1 (0.07, 0.11) | Ref. | Ref. | Ref. |
| Q2 (0.11, 0.14) | 0.51 (0.36, 0.66) <0.0001*** | 0.48 (0.33, 0.63) <0.0001*** | 0.43 (0.28, 0.58) <0.0001*** |
| Q3 (0.14, 0.16) | 0.43 (0.30, 0.56) <0.0001*** | 0.32 (0.19, 0.45) <0.0001*** | 0.27 (0.13, 0.40) 0.0002*** |
| Q4 (0.16, 6.09) | 0.90 (0.76, 1.04) <0.0001*** | 0.65 (0.51, 0.78) <0.0001*** | 0.62 (0.48, 0.76) <0.0001*** |
| **Pb quartile (µg/dL)** | | | |
| Q1 (0.05, 0.49) | Ref. | Ref. | Ref. |
| Q2 (0.49, 0.73) | −0.44 (−0.57, −0.32) <0.0001*** | −0.17 (−0.29, −0.05) 0.0070** | −0.21 (−0.33, −0.08) 0.0011** |
| Q3 (0.73, 1.15) | −0.70 (−0.83, −0.57) <0.0001*** | −0.30 (−0.44, −0.17) <0.0001*** | −0.36 (−0.50, −0.23) <0.0001*** |
| Q4 (1.15, 34.11) | −1.09 (−1.23, −0.95) <0.0001*** | −0.55 (−0.69, −0.41) <0.0001*** | −0.63 (−0.78, −0.48) <0.0001*** |
| **Hg quartile (µg/L)** | | | |
| Q1 (0.11, 0.20) | Ref. | Ref. | Ref. |
| Q2 (0.20, 0.34) | −0.34 (−0.57, −0.11) 0.0037** | −0.34 (−0.57, −0.12) 0.0029** | −0.30 (−0.52, −0.07) 0.0103* |
| Q3 (0.34, 0.62) | −0.09 (−0.33, 0.15) 0.4545 | −0.16 (−0.39, 0.08) 0.1875 | −0.12 (−0.35, 0.12) 0.3270 |
| Q4 (0.62, 28.84) | 0.00 (−0.24, 0.24) 0.9813 | −0.10 (−0.34, 0.13) 0.3916 | −0.05 (−0.29, 0.19) 0.6805 |

***: $P < 0.001$ **: $P < 0.01$ *: $P < 0.05$

Non-adjusted: without adjusting for covariates

Adjusted I: adjusting for gender, race/ethnicity, BMI and PIR

Adjusted II: adjusting for all covariates

contrast, PNR remained relatively stable at lower Hg concentrations and began to increase with higher Hg exposure. These findings indicate significant nonlinear relationships between Hg and the inflammatory ratios, with NLR, NMR, and PLR negatively correlated with Hg, whereas PNR displayed a positive correlation as Hg levels increased.

Subsequently, a log-likelihood ratio test was conducted to compare a single-line (non-segmented) model with a segmented regression model in order to determine the presence of an inflection point (Fig 3). The results presented in Table 3

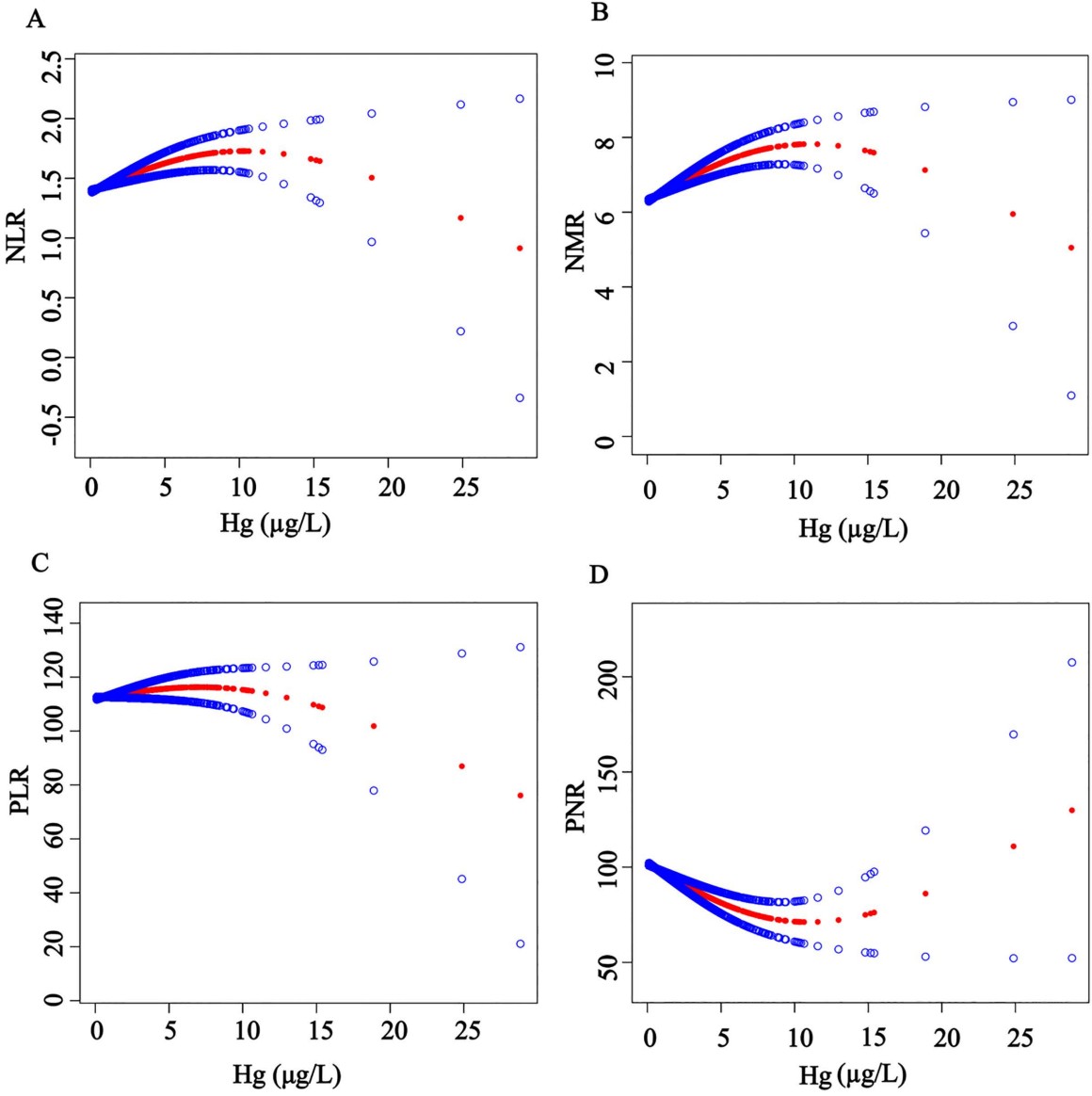

**Fig 3. Dose–response relationship between Hg and NLR, NMR, PLR and PNR.**

indicated that the *P*-values from the likelihood ratio tests were all below 0.05 (0.031, 0.012, 0.020, and < 0.001, respectively), suggesting that the two-piece linear regression model (Model 2) provided a significantly better fit than the standard linear regression model (Model 1). This findings suggest the existence of an inflection point in the relationship between Hg and the four inflammatory biomarkers. Specifically, NLR, NMR, and PNR showed statistically significant associations with Hg below the inflection point, whereas PLR demonstrated a significant association above the inflection point.

### 3.4. Subgroup analysis of age and BMI

The effects of Cd and Hg on NLR were statistically significant only in the teenager group, where each 1 μg/L increase in blood Cd was associated with a 0.18-unit increase in NLR, while a 1 μg/L increase in blood Hg corresponded to a

**Table 3. Two-piece wise and standard linear regression comparison.**

| Outcome (β, 95%CI, P value) | NLR | NMR | PLR | PNR |
|---|---|---|---|---|
| Model 1 Fitting model by standard linear regression | | | | |
| | 0.03 (0.02, 0.05) 0.0003 | 0.15 (0.10, 0.21) <0.0001 | −0.76 (−2.96, 1.45) 0.5040 | −3.09 (−4.21, −1.98) <0.0001 |
| Model 2 Fitting model by two-piece wise linear regression | | | | |
| Inflection point (K) | 10.23 | 12.53 | 14.8 | 10.4 |
| < K | 0.05 (0.02, 0.07) <0.0001 | 0.19 (0.12, 0.25) <0.0001 | 2.82 (−0.89, 6.54) 0.1403 | −3.95 (−5.21, −2.69) <0.0001 |
| > K | −0.04 (−0.12, 0.03) 0.2431 | −0.20 (−0.47, 0.08) 0.1668 | −6.20 (−11.28, −1.11) 0.0193 | 3.19 (−1.31, 7.69) 0.1653 |
| P for log likely-hood ratio test | 0.031 | 0.012 | 0.020 | <0.001 |

0.03-unit decrease in NLR. The influence of blood Cd on LMR exhibited different trends across age groups, showing a reduction of 2.85 units in LMR per 1 µg/L increase in blood Cd among toddlers, but an elevation of 1.15 units among school-age children. Similarly, increases in blood Cd were positively associated with NMR among school-age children (β = 1.14, 95% CI: 0.03–2.25) and teenagers (β = 0.36, 95% CI: 0.02–0.69). Except for the teenager group, increases in blood Cd concentration were positively associated with PLR across all other age groups, with decreasing β values as age increased: β = 79.26 (95% CI: 46.48–112.04), β = 78.64 (95% CI: 38.01–119.27), and β = 22.83 (95% CI: 5.85–39.81). In contrast, a significant association between blood Hg and PLR was observed only in the teenager group, showing a decrease of 2.29 units in PLR (β = −2.29, 95% CI: −3.72–−0.87). Additionally, blood Cd levels were negatively associated with PNR among teenagers but positively associated across the other three age groups. Finally, increases in blood Pb concentrations were positively associated with PNR across all age groups and with PLR among school-age children (Table 4).

Subgroup analyses by BMI category showed that an increases in blood Cd concentration were positively associated with elevated NLR across all BMI groups (Table 5). In contrast, higher blood Pb levels were negatively associated with NLR in both underweight (BMI < 25) and obese (BMI ≥ 30) individuals, with β values of −0.08 (95% CI: −0.10–−0.07) and −0.15 (95% CI: −0.28–−0.02), respectively. Blood Hg levels were positively associated with NLR in underweight individuals (β = 0.03, 95% CI: 0.01–0.05) but negatively associated in obese individuals (β = −0.15, 95% CI: −0.29–0.00). The relationships between heavy metals and LMR were most evident in underweight individuals, where both Cd and Hg were negatively associated with LMR (β = −0.98, 95% CI: −1.23–−0.74; β = −0.05, 95% CI: −0.09–0.00). In contrast, blood Pb levels were positively associated with LMR (β = 0.20, 95% CI: 0.17–0.24). Among individuals with BMI < 30, blood Cd levels were positively associated with NMR, with β values of 1.20 (95% CI: 0.89–1.52) and 0.68 (95% CI: 0.18–1.17) for specific BMI ranges. Conversely, Pb exposure was associated with a decrease in NMR. Both Cd and Hg were negatively associated with PNR in this group, with β values of −21.97 (95% CI: −28.17–−15.77) for Cd and −3.00 (95% CI: −4.16–−1.83) for Hg, showing progressively weaker negative associations as BMI increased. Additionally, Pb was positively associated with PNR across all BMI groups, with β values decreasing as BMI increased.

### 3.5. Mediation analysis of BMI

Fig 4 illustrates the statistically significant mediation effects of BMI as an intermediary variable linking Cd, Pb, and Hg with various inflammatory markers. For instance, BMI mediated 16% of the effect of blood Hg on LMR, whereas its mediation effects on the relationships between Hg and other inflammatory markers were not statistically significant (Fig 4I). Similarly, BMI mediated 30.39% of the effect of blood Cd on LMR (Fig 4B), 30.33% on NMR (Fig 4C), and 27.09% on NLR (Fig 4A), while no significant mediation were observed for Cd and the remaining inflammatory markers. In contrast, BMI exhibited

**Table 4. Stratified analysis of heavy metals and inflammatory markers in different age groups.**

|  | Age Group | Cd | Pb | Hg |
|---|---|---|---|---|
| NLR | Toddler (1 –3) | 0.48 (−0.02, 0.97) 0.0579 | −0.01 (−0.02, 0.01) 0.4271 | 0.01 (−0.02, 0.03) 0.5638 |
|  | Pre-School Child (4 –6 ) | −0.40 (−1.30, 0.50) 0.3799 | −0.03 (−0.07, 0.01) 0.1786 | −0.01 (−0.07, 0.05) 0.7701 |
|  | School-age Child (7–12) | −0.32 (−0.68, 0.04) 0.0818 | −0.01 (−0.05, 0.02) 0.4659 | −0.01 (−0.05, 0.02) 0.3720 |
|  | Teanager (1314151617–18) | 0.18 (0.07, 0.28) 0.0016* | −0.01 (−0.06, 0.03) 0.6089 | −0.03 (−0.07, −0.00) 0.0493* |
| LMR | Toddler (1–3) | −2.85 (−5.36, −0.35) 0.0257* | 0.01 (−0.05, 0.07) 0.7231 | 0.07 (−0.05, 0.19) 0.2579 |
|  | Pre-School Child (4–6) | 0.40 (−1.39, 2.18) 0.6634 | −0.00 (−0.08, 0.08) 0.9735 | 0.15 (0.02, 0.27) 0.0241* |
|  | School-age Child (7–12) | 1.15 (0.41, 1.90) 0.0024** | 0.00 (−0.07, 0.08) 0.9075 | 0.05 (−0.01, 0.12) 0.1189 |
|  | Teanager (13–18) | −0.06 (−0.24, 0.12) 0.5168 | 0.03 (−0.05, 0.10) 0.5127 | 0.02 (−0.03, 0.08) 0.3956 |
| NMR | Toddler (1–3) | 0.98 (−1.06, 3.02) 0.3477 | −0.04 (−0.09, 0.01) 0.1017 | 0.06 (−0.03, 0.16) 0.1988 |
|  | Pre-School Child (45–6) | 0.58 (−1.70, 2.86) 0.6203 | −0.07 (−0.18, 0.03) 0.1868 | 0.15 (−0.01, 0.32) 0.0613 |
|  | School-age Child (7–12) | 1.14 (0.03, 2.25) 0.0445* | −0.07 (−0.19, 0.05) 0.2301 | 0.06 (−0.04, 0.16) 0.2506 |
|  | Teanager (13–18) | 0.36 (0.02, 0.69) 0.0356* | −0.12 (−0.27, 0.02) 0.0884 | −0.08 (−0.19, 0.02) 0.1082 |
| PLR | Toddler (12–3) | 79.26 (46.48, 112.04) <0.0001*** | 0.44 (−0.39, 1.27) 0.2973 | −0.56 (−2.13, 1.01) 0.4871 |
|  | Pre-School Child (4–6) | 78.64 (38.01, 119.27) 0.0002*** | 1.88 (−0.00, 3.75) 0.0504 | −1.56 (−4.44, 1.33) 0.2902 |
|  | School-age Child (7891011–12) | 22.83 (5.85, 39.81) 0.0084** | 3.97 (2.17, 5.77) <0.0001*** | −1.36 (−2.91, 0.20) 0.0870 |
|  | Teanager (13–18) | −2.60 (−7.19, 1.99) 0.2671 | 0.63 (−1.35, 2.61) 0.5326 | −2.29 (−3.72, −0.87) 0.0016** |
| PNR | Toddler (12–3) | 96.92 (26.42, 167.41) 0.0071** | 3.03 (1.26, 4.80) 0.0008*** | −0.96 (−4.32, 2.41) 0.5784 |
|  | Pre-School Child (4–6) | 103.38 (59.06, 147.70) <0.0001*** | 3.00 (0.95, 5.05) 0.0042** | −0.89 (−4.04, 2.26) 0.5798 |
|  | School-age Child (7–12) | 45.96 (26.96, 64.97) <0.0001*** | 5.76 (3.75, 7.77) <0.0001*** | −1.29 (−3.03, 0.46) 0.1484 |
|  | Teanager (13–18) | −8.41 (−12.47, −4.35) <0.0001*** | 3.66 (1.91, 5.41) <0.0001*** | −0.33 (−1.59, 0.93) 0.6086 |

***: $P < 0.001$ **: $P < 0.01$ *: $P < 0.05$

**Table 5. Subgroup analysis of BMI groups.**

|  | BMI Group | Cd | Pb | Hg |
|---|---|---|---|---|
| NLR | < 25 | 0.50 (0.40, 0.61) <0.0001*** | −0.08 (−0.10, −0.07) <0.0001*** | 0.03 (0.01, 0.05) 0.0090** |
|  | 25 ≤ BMI < 30 | 0.24 (0.08, 0.41) 0.0044** | −0.04 (−0.14, 0.05) 0.3801 | 0.02 (−0.05, 0.08) 0.6264 |
|  | ≥ 30 | 0.40 (0.05, 0.75) 0.0246* | −0.15 (−0.28, −0.02) 0.0211* | −0.15 (−0.29, −0.00) 0.0446* |
| LMR | < 25 | −0.98 (−1.23, −0.74) <0.0001*** | 0.20 (0.17, 0.24) <0.0001*** | −0.05 (−0.09, −0.00) 0.0459* |
|  | 25 ≤ BMI < 30 | 0.09 (−0.21, 0.40) 0.5436 | 0.09 (−0.09, 0.27) 0.3276 | −0.04 (−0.16, 0.08) 0.5357 |
|  | ≥ 30 | −0.66 (−1.16, −0.17) 0.0086** | 0.21 (0.03, 0.39) 0.0219* | 0.08 (−0.12, 0.28) 0.4448 |
| NMR | < 25 | 1.20 (0.89, 1.52) <0.0001*** | −0.22 (−0.27, −0.17) <0.0001*** | 0.12 (0.06, 0.18) <0.0001*** |
|  | 25 ≤ BMI < 30 | 0.68 (0.18, 1.17) 0.0072** | −0.30 (−0.59, −0.01) 0.0408* | 0.01 (−0.18, 0.21) 0.8837 |
|  | ≥ 30 | −0.17 (−1.38, 1.04) 0.7866 | −0.15 (−0.59, 0.29) 0.4977 | −0.45 (−0.95, 0.04) 0.0725 |
| PLR | < 25 | 12.50 (7.51, 17.49) <0.0001*** | −1.73 (−2.51, −0.95) <0.0001*** | −0.10 (−1.03, 0.84) 0.8401 |
|  | 25 ≤ BMI < 30 | −4.05 (−11.39, 3.28) 0.2790 | 0.98 (−3.31, 5.28) 0.6542 | −2.77 (−5.67, 0.14) 0.0625 |
|  | ≥ 30 | 7.41 (−5.49, 20.32) 0.2604 | −2.44 (−7.15, 2.27) 0.3108 | −3.13 (−8.42, 2.16) 0.2471 |
| PNR | < 25 | −21.97 (−28.17, −15.77) <0.0001*** | 7.19 (6.23, 8.16) <0.0001*** | −3.00 (−4.16, −1.83) <0.0001*** |
|  | 25 ≤ BMI < 30 | −11.02 (−17.00, −5.05) 0.0003*** | 5.37 (1.87, 8.88) 0.0027** | −2.77 (−5.14, −0.39) 0.0227* |
|  | ≥ 30 | −3.30 (−13.03, 6.42) 0.5054 | 4.32 (0.79, 7.86) 0.0168* | 3.66 (−0.32, 7.63) 0.0721 |

***: $P < 0.001$ **: $P < 0.01$ *: $P < 0.05$

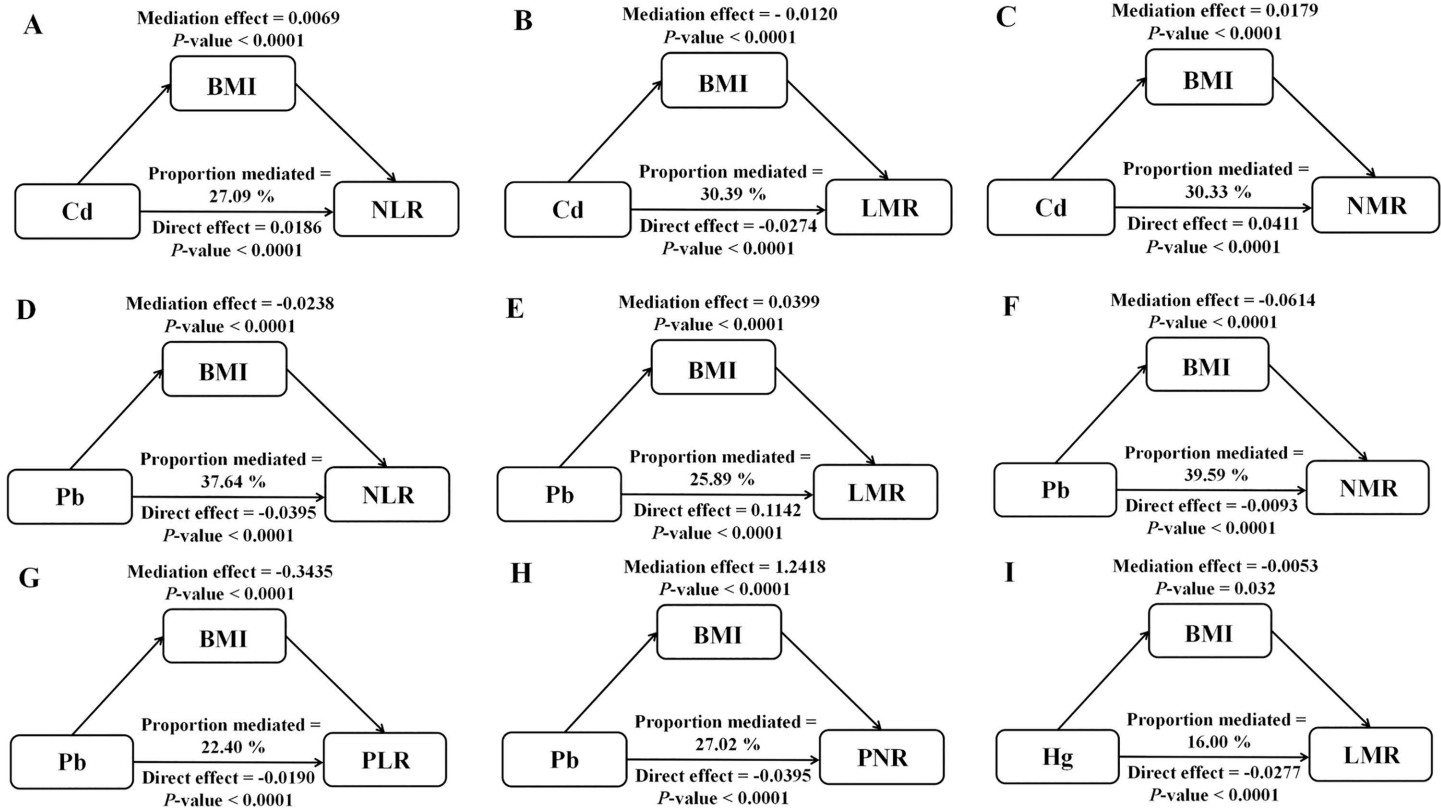

**Fig 4. Mediation analysis of heavy metals on biomarker levels via BMI.** Fig 4A-4C: mediation effects of BMI on the associations between Cd and NLR, LMR, and NMR; Fig 4D-4H: mediation effects of BMI on the associations between Pb and NLR, LMR, NMR, PLR and PNR; Fig 4I: mediation effects of BMI on the associations between Hg and LMR.

statistically significant mediation effects on all five inflammatory markers in relation to Pb, mediating 37.64% of the effect of blood Pb on NLR (Fig 4D), 22.40% on PLR (Fig 4G), 25.89% on LMR (Fig 4E), 27.02% on PNR (Fig 4H), and 39.59% on NMR (Fig 4F).

## 4. Discussion

This present study demonstrates significant association between the blood concentrations of Pb, Cd and Hg and various inflammatory ratios, with BMI serving as a key mediator role in these relationships. Specifically, Pb exposure is linked to reduction in NLR and PLR, whereas Cd exhibits an opposite pattern. These findings suggest that BMI mediates the effects of heavy metal exposure on inflammatory responses among minors, indicating that BMI is closely related to heavy metal exposure and may exacerbate health risks by influencing inflammatory responses.

Research suggests that heavy metals, as EDCs, can interfere with hormone-mediated processes that are essential for growth and development during pregnancy, infancy, and childhood. Such disruptions may increases the risk of overweight or obesity in minors [24–26]. Furthermore, because of their higher metabolic rates and greater exposure to certain EDCs due to developmental characteristics in behavior, anatomy, and physiology, children may be particularly vulnerable to environmental stressors, heavy metals and other EDCs [1,15]. Heavy metal exposure may impair the function of the growth hormone (GH) and insulin-like growth factor (IGF-1) axis, thereby affecting the growth and development of adolescents [27]. Disruption of thyroid function has also been reported, leading to alterations in thyroid-stimulating hormone (TSH) and

thyroid hormones levels [28]. For example, prenatal exposure to heavy metal in pregnant women has been shown to influence both maternal and neonatal thyroid hormones concentrations and iodine uptake, which may consequently interfere with the normal growth and development of minors and alter their basal metabolic rate, resulting in abnormal changes in weight and height [29]. Cd can mimic or interfere with hormonal signaling and promote fat accumulation by inducing insulin resistance, ultimately leading to an increase in BMI [30]. Beyond endocrine regulation, heavy metal exposure can also stimulate the over production of reactive oxygen species (ROS), resulting in oxidative stress. Elevated oxidative stress markers in children promotes adipogenesis and enhance lipid accumulation, further contributing to BMI elevation [31,32]. Blood Pb levels can damage the gastrointestinal mucosa, impairing its barrier function and disrupting absorption of essential nutrients such as iron and calcium, thereby affecting children's growth and body weight [33,34]. These mechanisms collectively contribute to the observed association between heavy metal exposure and increased BMI. Heavy metals interfere with endocrine and metabolic pathways, promoting weight gain and predisposing individuals to enhanced inflammatory responses. Excess adipose tissue further induces chronic low-grade inflammation, which adversely affects immune system function in obese children [35]. Obesity is accompanied by elevated levels of pro-inflammatory cytokines such as TNF-α and IL-6, both of which are closely associated with insulin resistance and metabolic disorders, as well as diminished immune cell activity [36]. Consequently, obese children are more vulnerable to metabolic dysregulation and exhibit weakened immune defenses [37].

Based on the above evidence, we hypothesize that heavy metal exposure contributes to increased BMI, which in turn affects immune function and alters inflammatory ratios. This mediating effect is particularly relevant in minors, whose metabolic and immune systems are still developing, making them more susceptible to these influences. Accordingly, the present study investigates the mediating role of BMI in the association between blood concentrations of Pb, Cd, and Hg and inflammation-related hematological ratios in children. Blood Pb concentration was positively associated with LMR and PNR, with 25.89% and 27.02% of these effects mediated by BMI, respectively. In contrast, Pb was negatively associated with NLR, PLR, and NMR, with corresponding BMI-mediated proportions of 37.64%, 22.40%, and 39.59%. Blood Cd concentration was positively associated with NLR and NMR, with BMI mediating 27.09% and 30.33% of these effects, respectively, while Cd showed a negative association with LMR and a BMI-mediated proportion of 30.39%. Additionally, blood Hg concentration was negatively associated with LMR, with 16.00% of its effect mediated by BMI. These findings are consistent with previous studies. He et al. reported that overweight and obese school-aged girls exhibited significantly elevated leukocyte and neutrophil counts, which were positively correlated with insulin resistance (HOMA-IR). These indicators may serve as potential biomarkers of insulin resistance, underscoring the pivotal role of obesity-induced chronic inflammation in the development of metabolic disorders [38]. Similarly, another study found that overweight or obese children showed increased leukocyte counts and NLR, which were strongly associated with insulin resistance in boys. Collectively, these results suggest that inflammatory ratio, such as leukocyte counts and NLR, may serve as early predictors of obesity-related metabolic abnormalities in children [39]. This evidence further supports the notion that BMI function as a key linking heavy metal exposure to inflammatory responses. Understanding this mechanism is essential for developing health interventions aimed at adolescents.

This study elucidates the intricate relationship between heavy metal exposure, BMI, and inflammatory responses in minors. The combined adverse effects of heavy metal exposure and increased BMI on the adolescent immune system may predispose individuals to long-term health complications, including metabolic syndrome and cardiovascular diseases. Therefore, public health authorities should strengthen surveillance of environmental heavy metal pollution, particularly in regions with high exposure risks among children. Implementing stricter environmental regulations, enhancing early screening for vulnerable populations, and promoting healthy lifestyle habits may collectively help mitigate obesity and reduce the burden of inflammation-related diseases in children.

Although this study underscores the significant mediating role of BMI in the association between heavy metal exposure and inflammatory responses, its cross-sectional design limits the ability to infer causal relationships. Future research

should employ longitudinal approaches to elucidate the long-term effects of heavy metal exposure on BMI dynamics and inflammatory profiles. Moreover, incorporating additional potential confounders—such as dietary patterns, physical activity, socioeconomic factors, and genetic polymorphisms—would enable a more comprehensively understanding of these complex interactions. Further mechanistic studies are warranted to clarify how heavy metals influence metabolic and immune pathways in minors, thereby providing a stronger scientific foundation for targeted prevention and intervention strategies.

## 5 Conclusion

This study demonstrates that BMI acts as a significant mediator between blood heavy metal concentrations and inflammatory indicators in minors, offering a novel perspective on the health impacts of heavy metal exposure during adolescence. These findings suggests that targeted public health policies should be implemented to minimize heavy metal exposure in minors, prevent obesity, and mitigate the risk of metabolic and inflammation-related disorders in this vulnerable population.

## Acknowledgments

The authors express their gratitude to the staff members for their valuable contribution to data collection and for their efforts in making the data publicly accessible.

## Author contributions

**Conceptualization:** Xinpeng Li.

**Data curation:** Lu Han.

**Methodology:** Xinpeng Li, Lu Han.

**Software:** Xinpeng Li, Lu Han.

**Writing – original draft:** Xinpeng Li, Lu Han.

**Writing – review & editing:** Xinpeng Li.

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
