## [Decision Letter · Decision Letter 0]

7 Oct 2025

Dear Dr. Han,

Thank you for submitting your manuscript to PLOS ONE. After careful consideration, we feel that it has merit but does not fully meet PLOS ONE’s publication criteria as it currently stands. Therefore, we invite you to submit a revised version of the manuscript that addresses the points raised during the review process.

Both reviewer reports recommended **minor revision** , and their comments were consistent in suggesting relatively small but important changes. Therefore, we invite you to submit a **revised version**

A rebuttal letter that responds to each point raised by the academic editor and reviewer(s), uploaded as a separate file labeled Response to Reviewers.A marked-up copy of your manuscript with changes highlighted, labeled Revised Manuscript with Track Changes.A clean version of your revised manuscript without tracked changes, labeled Manuscript.

We look forward to receiving your revised manuscript.

Hiroyoshi Iwata, MD, MSc, PhD

Academic Editor

PLOS ONE

2. In the online submission form, you indicated that [The data supporting the findings of this study are available on the NHANES Website (https://www.cdc.gov/nchs/nhanes) or contact the corresponding author.].

Additional Editor Comments:

Decision on Manuscript: Heavy Metal Exposure and Its Impact on Inflammatory Ratios in Minors: The Mediating Role of BMI

Both reviewer reports recommended minor revision, and their comments were consistent in suggesting relatively small but important changes.

In light of this, the editorial decision is to invite you to submit a revised version of your manuscript.

Please carefully address each of the reviewers’ comments in your revision and provide a point-by-point response.

We look forward to receiving your revised manuscript.

Best regards,

Hiroyoshi Iwata, MD, PhD

Academic Editor

PLOS ONE

Reviewers' comments:

Reviewer's Responses to Questions

**Comments to the Author**

1. Is the manuscript technically sound, and do the data support the conclusions?

Reviewer #1: Yes

Reviewer #2: Yes

2. Has the statistical analysis been performed appropriately and rigorously?

Reviewer #1: Yes

Reviewer #2: Yes

3. Have the authors made all data underlying the findings in their manuscript fully available?

Reviewer #1: Yes

Reviewer #2: Yes

4. Is the manuscript presented in an intelligible fashion and written in standard English?

Reviewer #1: Yes

Reviewer #2: Yes

Reviewer #1: The manuscript “Heavy Metal Exposure and Its Impact on Inflammatory Ratios in Minors: The Mediating Role of BMI” offers insight into the effect of heavy metals on the health of the minors which has been exclusively studied in adults till now, imparting novelty to the present work. But the manuscript needs certain modifications to enhance its suitability for publication in this journal, which are listed below:

1. Manuscript needs to be checked by native English speaker to negate any language related errors, in result section, prefer to keep the same verb tense which is fluctuating from past to simple present tense.

2. The abbreviations for neutrophil-to-lymphocyte ratio, lymphocyte-to-monocyte ratio, platelet-to-lymphocyte ratio, platelet-to-neutrophil ratio, and neutrophil-to-monocyte ratio have been mentioned in introduction section. Do not specify the same in section 2.3 (line no 102).

3.Include the significance of the inflammatory markers for better understanding of the relationship between heavy metals and inflammatory markers.

Reviewer #2: This study explores the relationship between heavy metal exposure and inflammatory ratios in minors, with BMI as a potential mediator, using NHANES 2007–2018 data. The large sample size and age-group stratification strengthen the findings. The use of linear regression and mediation analysis is appropriate and reveals consistent patterns: blood Pb, Cd, and Hg are significantly associated with inflammatory ratios, and BMI mediates 20–40% of these effects. The results are clearly presented and the conclusions are supported by the data. I recommend that the authors expand the methodology and figure legends.

**Do you want your identity to be public for this peer review?** For information about this choice, including consent withdrawal, please see our Privacy Policy

Reviewer #1: No

Reviewer #2: No

---

## [Author Response · Author response to Decision Letter 1]

1 Nov 2025

Response to Reviewer 1:

Reviewer #1: The manuscript “Heavy Metal Exposure and Its Impact on Inflammatory Ratios in Minors: The Mediating Role of BMI” offers insight into the effect of heavy metals on the health of the minors which has been exclusively studied in adults till now, imparting novelty to the present work. But the manuscript needs certain modifications to enhance its suitability for publication in this journal, which are listed below:

1.Manuscript needs to be checked by native English speaker to negate any language related errors, in result section, prefer to keep the same verb tense which is fluctuating from past to simple present tense.

Response 1: We sincerely appreciate the reviewer’s valuable suggestion. In response, we have carefully refined the language throughout the manuscript to improve clarity and precision without altering the original meaning. Furthermore, the verbs in the Methods and Results sections have been consistently revised to the past tense where appropriate. All these revisions are clearly highlighted in the tracked changes of the revised manuscript.

2.The abbreviations for neutrophil-to-lymphocyte ratio, lymphocyte-to-monocyte ratio, platelet-to-lymphocyte ratio, platelet-to-neutrophil ratio, and neutrophil-to-monocyte ratio have been mentioned in introduction section. Do not specify the same in section 2.3 (line no 102).

Response 2: We sincerely thank the reviewer for the insightful comment. The abbreviations PC, NC, LC, and MC in Section 2.3 (line 102) refer to platelet count, neutrophil count, lymphocyte count, and monocyte count, respectively. These abbreviations differ from those defined earlier in the Introduction (i.e., NLR, LMR, PLR, PNR, and NMR), which represent ratio-based indicators. As PC, NC, LC, and MC appear for the first time in this section, we have retained their definitions here for clarity.

3.Include the significance of the inflammatory markers for better understanding of the relationship between heavy metals and inflammatory markers.

Response 3: We have added a description in lines 72–78 of the Introduction section, emphasizing the close association between heavy metal exposure, disruption of immune homeostasis, and the promotion of systemic inflammation. This revision helps readers better understand the relationship between heavy metals and inflammatory responses.

Response to Reviewer 2:

Reviewer #2: This study explores the relationship between heavy metal exposure and inflammatory ratios in minors, with BMI as a potential mediator, using NHANES 2007–2018 data. The large sample size and age-group stratification strengthen the findings. The use of linear regression and mediation analysis is appropriate and reveals consistent patterns: blood Pb, Cd, and Hg are significantly associated with inflammatory ratios, and BMI mediates 20–40% of these effects. The results are clearly presented and the conclusions are supported by the data. I recommend that the authors expand the methodology and figure legends.

Response: We have added a detailed description of the mediation effect in Section 2.6 “Mediation analysis” of the Methods, lines 135-147. In addition, we have revised and enriched the contents to provide clearer explanations of variables, models, and significance indicators in Figure 4 (Section 3.5, lines 246–256). The legend for Figure 4 has also been made more detailed to improve clarity, lines 257-260.

---

## [Decision Letter · Decision Letter 1]

8 Dec 2025

Heavy Metal Exposure and Its Impact on Inflammatory Ratios in Minors: The Mediating Role of BMI

PONE-D-25-21195R1

Dear Dr. Han,

We’re pleased to inform you that your manuscript has been judged scientifically suitable for publication and will be formally accepted for publication once it meets all outstanding technical requirements.

Kind regards,

Hiroyoshi Iwata, MD, MSc, PhD, FACP

Academic Editor

PLOS One

Reviewers' comments:

Reviewer's Responses to Questions

**Comments to the Author**

Reviewer #1: All comments have been addressed

Reviewer #2: All comments have been addressed

2. Is the manuscript technically sound, and do the data support the conclusions?

Reviewer #1: Yes

Reviewer #2: Yes

3. Has the statistical analysis been performed appropriately and rigorously?

Reviewer #1: Yes

Reviewer #2: Yes

4. Have the authors made all data underlying the findings in their manuscript fully available?

Reviewer #1: Yes

Reviewer #2: Yes

5. Is the manuscript presented in an intelligible fashion and written in standard English?

Reviewer #1: Yes

Reviewer #2: Yes

Reviewer #1: The manuscript has been revised with all comments being addressed; therefore, it is now suitable for publication.

Reviewer #2: Using adult BMI cutoffs for children can lead to misclassification and may substantially affect the accuracy of the study’s interpretations, as pediatric BMI must be assessed using age- and sex-specific percentiles rather than adult thresholds.

**Do you want your identity to be public for this peer review?** For information about this choice, including consent withdrawal, please see our Privacy Policy

Reviewer #1: No

Reviewer #2: No

---

## [Editor Report · Acceptance letter]

PONE-D-25-21195R1

PLOS One

Dear Dr. Han,

I'm pleased to inform you that your manuscript has been deemed suitable for publication in PLOS One. Congratulations! Your manuscript is now being handed over to our production team.

Kind regards,

on behalf of

Dr. Hiroyoshi Iwata

Academic Editor

PLOS One